# Risk factors, ethnicity and dementia: A UK Biobank prospective cohort study of White, South Asian and Black participants

**Naaheed Mukadam**[1,2]*, **Louise Marston**[1], **Gemma Lewis**[1], **Gill Livingston**[1,2]

**1** Division of Psychiatry, University College London, London, Greater London, United Kingdom, **2** Camden and Islington NHS Foundation Trust, London, Greater London, United Kingdom

* n.mukadam@ucl.ac.uk

## Abstract

### Background

Our knowledge of the effect of potentially modifiable risks factors on people developing dementia is mostly from European origin populations. We aimed to explore if these risk factors had similar effects in United Kingdom (UK) White, South Asian and Black UK Biobank participants recruited from 2006–2010 and followed up until 2020.

### Methods

We reviewed the literature to 25.09.2020 for meta-analyses identifying potentially modifiable risk factors preceding dementia diagnosis by ≥10 years. We calculated prevalence of each identified risk factor and association with dementia for participants aged ≥55 at registration in UK Biobank. We calculated hazard ratios using Cox regression for each risk factor, stratified by ethnic group, and tested for differences using interaction effects between each risk factor and ethnicity.

### Findings

We included education, hearing loss, hypertension, obesity, excess alcohol consumption, physical inactivity, smoking, high total cholesterol, depression, diabetes, social isolation, and air pollution as risks. Out of 294,162 participants, there were 287,806 White, 3590 South Asian and 2766 Black people, followed up for up to 14.8 years, with a total follow-up time of 3,392,095 years. During follow-up, 5,972 people (2.03%) developed dementia. Risk of dementia was higher in Black participants than White participants (HR for dementia compared to White participants as reference 1.43, 95% CI 1.16–1.77, p = 0.001) but South Asians had a similar risk. Association between each risk factor and dementia was similar in each ethnic group with no evidence to support any differences.

### Interpretation

We find that Black participants were more likely to develop dementia than White participants, but South Asians were not. Identified risk factors in White European origin

**Data Availability Statement:** Data cannot be shared publicly due to restrictions from UK Biobank. Others can access these data by applying to UK Biobank (https://www.ukbiobank.ac.uk/

enable-your-research/apply-for-access). The authors had no special privileges with regards to data access.

**Funding:** NM is funded by an Alzheimer's Society Senior Research Fellowship (AS-SF-18b-001). All authors are supported by the UCLH NIHR Biomedical Research Centre. The funders had no role in study design, data collection and analysis, decision to publish, or preparation of the manuscript.

**Competing interests:** The authors have declared that no competing interests exist.

participants had a similar effect in Black and South Asian origin participants. Volunteers in UK Biobank are not representative of the population and interaction effects were underpowered so further work is needed.

## Background

The number of people living with dementia worldwide is predicted to rise from 46 to >131 million between 2015 and 2050 [1]. Despite around 60% of people with dementia living in low- and middle-income countries [1], and prevalence of dementia varying between some ethnic groups within countries [2, 3], most of our understanding of dementia is based on studies in populations of European descent [2]. A number of potentially modifiable risk factors have been identified for all-cause dementia worldwide [2, 3] (rather than dementia subtypes) but again, most evidence for these is from people of European origin living in higher income countries. There are some data on differing frequencies of dementia risk factors in minority ethnic groups compared to majority populations [4, 5]. However, to our knowledge there is no published research on the association between these risk factors and dementia in diverse ethnic groups and this information is crucial in order to target prevention and understand disease mechanisms.

The UK has a sizeable minority ethnic population (around 13%), of whom the largest non-White group is South Asian and the second largest is Black (African or Caribbean) [6]. One risk factor for dementia, diabetes, is more prevalent in South Asian countries [7] and in South Asians in the UK [4, 5] compared to European-origin populations. Two other risk factors, hypertension [8] and obesity [9], while less common in South Asia than in Europe, are increasing in South Asia and are more common in South Asians in the UK than in their countries of origin [10, 11]. Moreover, in South Asians with hypertension, the impact of blood pressure on the risk of stroke is higher than in their White British counterparts [10], so there is a much larger effect at population level.

Diabetes, stroke and hypertension are more common in people of African Caribbean origin in the UK compared to White British people [5] and African Caribbean people with diabetes were previously found to be six times more likely to die from stroke compared to Europeans with diabetes [12]. Dementia is more common in British African Caribbean people compared to White British people and onset of the disease is at a younger age [13]. Compared with European populations, those of African descent have higher rates of hypertension, diabetes and obesity, but smoking rate and alcohol consumption is lower [14]. Mean education levels are also lower in South Asia and Africa than in Europe which could be relevant for dementia risk in people who have migrated from these regions (https://ourworldindata.org/global-education#all-charts-preview).

We aimed to address a critical need to understand risks for all-cause dementia across ethnic groups, and whether ethnicity affects the association between established risk factors and dementia risk, in order to increase knowledge about the cause and prevention of dementia in these minority ethnic groups.

## Method

We registered the protocol for this project online prior to data analysis (https://osf.io/d2j4z/).

## Sample

The UK Biobank is a cohort of more than 500,000 participants, with varying socioeconomic and ethnic characteristics [15], all aged between 40 and 69 at registration from 2006 to 2010. Participants completed demographic and health questionnaires, interviews and blood, urine and saliva tests. UK Biobank also links participants to their existing health records, such as those from general practice (GP), hospitals and those collected centrally (cancer and death statistics for instance). The UK Biobank received approval from the National Information Governance Board for Health and Social Care and the National Health Service North West Multicentre Research Ethics Committee. All participants provided informed consent through electronic signature at baseline assessment.

We only included participants aged 55 and over at baseline, as younger participants would have a very small risk of developing dementia by the end of the approximately 15-year follow-up period.

## Identification of risk factors

To identify risk factors to include in the analysis, we searched Medline, Embase and Allied and Complementary Medicine databases from inception until 25[th] September 2020 for systematic reviews of dementia risk factors using search terms "review" OR "meta-analysis" AND "risk factors" AND "dementia" OR "AD" with no limit on start date or language. We pre-specified that we would include data from meta-analyses about exactly specified (not "hormones" for example); risk factors which preceded dementia diagnosis by 10 or more years (to reduce the risk of reverse causality); evidence which indicated whether the risk factors were relevant in early, mid or later life and used specified clinical criteria to define risk factors. We excluded risk factors that could be part of the dementia syndrome or diagnosis such as motor symptoms related to Parkinson's disease or stroke. NM reviewed the retrieved papers to see if they fulfilled the inclusion criteria.

## Analysis of association of risk factors with dementia

**Ethnicity.** All participants self-identified their ethnicity using the UK Census categories. For this study we combined ethnicities into White (British and Irish); South Asian (Bangladeshi, Indian, Pakistani); or Black or Black British (Black African, Black Caribbean, Black British, Black Other), as low frequency genetic variants cluster in people originating in the same geographic location [16]. All other ethnic groups were grouped into a fourth ethnic group and included in analyses but they were heterogeneous and were not part of our main hypotheses. We included only those with a recorded valid ethnicity in our analyses as this was our main exposure variable of interest.

**Main outcome.** Diagnosis of dementia was established through self-report or linked electronic health records in the UK Biobank using a defined algorithm and recognised ICD9 and ICD10 codes [17]. The positive predictive values (PPVs) for the diagnosis of all-cause dementia were 86.8% for primary care, 87.3% for hospital admissions and 80.0% for mortality data respectively and 82.5% across all datasets [18] when compared against blinded clinician diagnosis from notes using standardised diagnostic criteria. Subtypes of dementia were determined to be much less accurate. This indicates that all-cause dementia as identified within UK Biobank is a valid diagnosis. We excluded people with prevalent dementia reported at or before their baseline assessment.

**Covariates.** We considered the following variables based on data, which participants gave at baseline, as they might influence the association between ethnicity, and dementia.

Age—We adjusted for age at baseline in years in all models.

Sex–participants self-identified as male or female and we adjusted for sex in all models.

Townsend score–this is a marker of area-level socioeconomic deprivation and was based on the census output area (minimum 40 households) of participants' residential postcodes [19]. A higher score equates to more deprivation. The mean population score is zero, so any area above zero is deprived and below zero is more affluent [20]. We adjusted all models for Townsend score.

We did not include a marker for general health as a covariate because: 1) we wanted to directly compare estimates for all risk factors to previous work which usually adjusted only for age and sex [2] and 2) comorbidities may be on the causal pathway from ethnicity to dementia, for most of the variables under consideration, especially cardiovascular risk factors and therefore including them would bias the effect estimate.

**Statistical analysis.** We described demographics and frequencies of each risk factor across the White, South Asian and Black groups. We also calculated crude incidence rates of dementia separately for each ethnic group.

Start date of risk was determined as the date participants joined the study. For purposes of determining follow-up time, end date was the earliest of: first date of recorded dementia diagnosis; date lost to follow-up; date of death; end of study (18th December 2020 –as this was the last date on which we obtained data for this study). The latter two were time points for censoring the data. Our primary analysis was for those with complete data for ethnicity, risk factors and all covariates.

We conducted a Cox regression using ethnicity as the main exposure and all cause dementia as the main outcome, before and after adjusting for age, sex and Townsend score. We tested the proportionality of hazards assumption using Schoenfeld residuals. To explore this association further we conducted post-hoc analyses, adding other risk factors from our analyses to the model in turn to investigate whether the association between Black ethnicity and dementia was attenuated or eliminated after adjusting for other known modifiable risk factors for dementia. To investigate potential genetic contribution to increased dementia risk in people of self-defined Black ethnicity, we used two single nucleotide polymorphisms (SNPs) which are well established markers for the APOE gene (rs429358 and rs7412) [21] to categorise participants into low (one or two ε2 alleles), intermediate (ε3), higher (one ε4) and high (two ε4 alleles) genetic risk. These markers were provided as part of our requested dataset for another planned analysis and have been subjected to UK Biobank quality control procedures [22].

To investigate whether ethnicity modifies the relationship between modifiable risk factors and dementia, we ran Cox regression models for dementia and each risk factor separately, including the risk factor itself, age, sex and Townsend score, stratified by the three ethnic groups under consideration. We also ran Cox regression models for each risk factor, including an interaction term for ethnicity (White British vs other ethnic groups) and the risk factor.

## Results

### Risk factors

In the literature review, we found consistent and strong associations between 12 risk factors which were included in UK Biobank, and dementia over 10 year later. We include details on papers retrieved, lists of excluded risk factors with reasons for exclusion and risk factors that were not available in UK Biobank in the S1 File.

**Exposure variables/Dementia risk factors.** We included the following risk factors: education, hypertension, obesity, hearing loss, diabetes, depression, social isolation, physical inactivity, diabetes, air pollution (particulate matter $\leq 2.5 \mu g/m^3$ –$PM_{2.5}$), raised cholesterol and

**Table 1. Risk factors and their definitions.**

| Risk factor | How measured in UK Biobank | Use in analysis |
|---|---|---|
| Education | Self-reported age at leaving education and highest qualification | Using both variables we categorised participants into two categories (up to age 16 and higher than this) [2] |
| Hypertension | Self-reported diagnosis or taking antihypertensive medication | Either self-report or reported use of medications was considered as evidence of hypertension [23] |
| Hearing loss | Self-reported problems with hearing including in a noisy environment; use of hearing aid | Positive responses to any of the questions were considered evidence of hearing impairment [24, 25] |
| Obesity | Weight and height were measured at baseline | We calculated Body Mass Index (BMI) using baseline measurements and defined obesity as a BMI>30kg/m$^2$ [26] |
| Excessive alcohol intake | Self-reported drinking status; frequency and quantity and types of alcohol consumed per week | Total units per week were calculated using frequency*quantity. We categorised into those drinking no alcohol, ≤21 or over 21 units per week [2, 27, 28] |
| Smoking | Self-reported status as current, previous or never smoker | We categorised participants into current and non-current smokers [29] |
| Depression | Participants were asked if they had ever seen their doctor for anxiety or depression | Those who answered yes were categorised as having a history of depression which showed a modest correlation with algorithmically derived depression validated in a subset of participants [30] |
| Social isolation | Cohabitation; self-reported frequency of contact with friends and family | Answers were used to categorise participants into those with daily/almost daily social contact and less frequent contact [31, 32] |
| Physical inactivity | Self-reported physical activity over previous four weeks–duration, intensity and frequency | We used published guidance to calculate metabolic equivalents (MET* minutes) per week based on published guidance [33]. We categorised participants into those meeting World Health Organisation guidance for physical activity [26] and those not. |
| Air pollution | Air pollution estimates for the year 2010 were modelled for each address using a Land Use Regression (LUR) model developed as part of the European Study of Cohorts for Air Pollution Effects (ESCAPE) (http://www.escapeproject.eu/) | The best evidence for dementia risk is for PM$_{2.5}$ air pollution by [34] so we used this as our exposure. We classified this into a binary variable, splitting into concentrations below or equal to and above the World Health Organisation recommended threshold [35] of an annual average of 10μg/m$^3$ |
| Diabetes | Self-reported diagnosis; use of diabetes medication | A positive response to either question was considered as evidence of diabetes [36] |
| Total cholesterol | Blood sample at baseline | We categorised participants into those with total cholesterol within normal limits or high (>6.5mmol/l) in line with the usual classification [37] |

*MET = Metabolic equivalent of task

excessive alcohol consumption. Table 1 defines these variables and explains how we operationalised them.

## Completeness of data

There was no missing data on age, sex, hypertension, hearing loss, obesity, physical inactivity and alcohol intake. Variables with missing data were smoking (0.59%), Townsend score (0.13%), social isolation (0.20%), diabetes (0.52%), educational attainment (0.74%), ethnicity (0.74%), depressive symptoms (1.0%), cholesterol (6.56%) and air pollution (8.22%). Participants with any missing data and complete data were similar in terms of mean age, sex, education, ethnicity, Townsend score, length of follow up and dementia cases (see Table 2).

## Characteristics of sample

We included 294,162 people aged 55 and over who were followed up for up to 14.8 years, with a total follow up time of 3,392,095 person years. There were 5,972 incident diagnoses of dementia. We included 287,806 White, 3,590 South Asian and 2,766 Black participants. Characteristics of the sample according to ethnic groups are presented in Table 3. All groups were

**Table 2. Demographic characteristics in those with and without any missing data.**

|  | No missing data | Missing data |
|---|---|---|
|  | N = 248,710 | N = 45,452 |
| **Age—mean(SD)** | 62 (4) | 62(4) |
| **Sex (% female)** | 53.4 | 55.2 |
| **Age at leaving full-time education–mean (SD)** | 18(3) | 18(4) |
| **White (%)** | 97.9 | 97.6 |
| **South Asian (%)** | 1.2 | 1.4 |
| **Black (%)** | 0.9 | 1.0 |
| **Townsend score–mean (SD)** | -1.6(2.9) | -1.1(3.3) |
| **Follow up time in years–mean (SD)** | 11.5(1.8) | 11.8(2.3) |
| **Dementia diagnosis (%)** | 1.9 | 2.5 |

similar in age and there was a greater percentage of female participants in the White and Black compared to South Asian groups. South Asians had the highest frequency of higher levels of education, followed by Black and then White participants. South Asian and Black participants had higher mean levels of deprivation, higher levels of hypertension, inactivity, exposure to air pollution and diabetes compared to the White participants. Black participants had the highest frequency of obesity and hypertension. Both South Asian and Black participants were less likely to smoke, have high cholesterol, depression and consume excess alcohol compared to White participants. South Asians were least likely to experience low frequency of social contact, followed by White participants and Black people had less frequent social contact. In those aged 55 and over, 2.8% of White, 1.3% of South Asian and 6.7% of Black participants had two ε4 alleles and 27.8% of White, 17.5% of South Asian and 36.3% of Black participants had one ε4 allele.

**Table 3. Characteristics of participants in different ethnic groups *PYAR = person years at risk.**

| Characteristic | White (N = 287,806) | South Asian (N = 3,590) | Black (N = 2760) |
|---|---|---|---|
| **Age—mean (SD)** | 62(4) | 62(4) | 61(5) |
| **% Female** | 53.5 | 45.9 | 58.2 |
| **Townsend score mean (SD)** | -1.6(2.9) | 0.0(3.0) | 2.5(3.5) |
| **Education up to age 16 only N(%)** | 144051(50.2) | 711(20.5) | 898(33.2) |
| **Hypertension N(%)** | 103173(35.9) | 1713(47.7) | 1594(57.6) |
| **Hearing loss N(%)** | 130004(45.2) | 1675(46.7) | 869(31.4) |
| **Obesity N(%)** | 73148(25.4) | 932(26.0) | 1165(42.1) |
| **Drinking >21 units alcohol per week N(%)** | 57241 (19.9) | 213(5.9) | 171(6.2) |
| **Current smoker N(%)** | 25398(8.9) | 235(6.6) | 224(8.2) |
| **History of depression N(%)** | 96346 (33.7) | 752(21.7) | 549(20.2) |
| **Social contact less than daily/almost daily N(%)** | 48393(16.8) | 390(10.9) | 832(30.4) |
| **Inactivity N(%)** | 31631(11.0) | 705(19.6) | 489(17.7) |
| **Air pollution >WHO recommendation N(%)** | 116291(44.0) | 2016(57.9) | 2023(74.7) |
| **Diabetes N(%)** | 17586 (6.1) | 918(26.1) | 516(18.8) |
| **High total cholesterol N(%)** | 70293(26.1) | 405(12.3) | 352(14.0) |
| **APOE ε4 –one allele** | 80010(27.8) | 628(17.5) | 1001(36.3) |
| **APOE ε4 –two alleles** | 8058(2.8) | 47(1.3) | 185(6.7) |
| **Crude dementia incidence rate per 1000 PYAR*(95% CI)** | 1.75 (1.70–1.79) | 1.97(1.58–2.45) | 2.97(2.42–3.64) |
| **Median follow up time in years (IQR)** | 11.8(1.4) | 11.2(1.1) | 11.1(1.2) |

## Association between ethnicity and dementia

5,972 people developed dementia of whom 5,789 were White, 79 were South Asian, 91 were Black and 13 were from other ethnic groups. The crude incidence rates of dementia were similar for the White and South Asian participants but were higher for Black participants, as shown in Table 3. With White participants as the reference category, after adjusting for age, sex and Townsend score, there was no evidence that hazard ratios for dementia in South Asians were higher than in White participants (HR 1.09, 95% CI 0.88–1.37, p = 0.407) but those for Black participants were (HR 1.43, 95% CI 1.16–1.77, p = 0.001). Black ethnicity was associated with an increased risk of dementia compared to White participants even after all modifiable risk factors were adjusted for (HR 1.34, 95% CI 1.06–1.69, p = 0.014).

When we included APOE in the Cox regression model with all other risk factors the HR for dementia in Black participants was 1.04 (95%CI 0.80–1.36, p = 0.758) and in South Asians was 1.03 (95%CI 0.78–1.36, p = 0.836).

## Association between risk factors and dementia across ethnic groups

Table 4 shows the hazard ratios for dementia for each of the 12 risk factors, stratified by ethnic group and the p value for interaction terms in the regression models. Unadjusted Cox regression models are shown in the S1 File.

Higher levels of education were associated with a reduced risk of dementia in all ethnic groups, although confidence intervals were wide in the South Asian and Black groups but there was no evidence to support an interaction effect. Hypertension, hearing loss, obesity, smoking, social isolation, air pollution, depression and diabetes were all associated with an increase in dementia risk in White participants. Point estimates for these risk factors were similar in South Asian and Black participants although again, confidence intervals were wide in the minority ethnic groups and there was no evidence to support an interaction effect. Using moderate alcohol consumption as the reference group, consumption of no alcohol compared to moderate drinking (1–21 units/week) was associated with an increased risk of dementia in the White group (HR 1.42, 95% CI 1.34–1.50) with similar estimates in the South Asian (HR 1.59, 95% CI 0.93–2.71) and Black groups (HR 1.35, 95% CI 0.80–2.28). We found no evidence of an association with dementia in those drinking more than 21 units of alcohol when

**Table 4. Hazard ratios for each risk factor stratified by ethnic groups, adjusted for age, sex and Townsend score.**

| Risk factor | White HR | 95% CI Lower | 95% CI Upper | South Asian HR | 95% CI Lower | 95% CI Upper | Black HR | 95% CI Lower | 95% CI Upper | Interaction p-value |
|---|---|---|---|---|---|---|---|---|---|---|
| Education (above GCSE vs below) | 0.80 | 0.76 | 0.84 | 0.78 | 0.45 | 1.32 | 0.86 | 0.56 | 1.32 | 0.78 |
| Hypertension | 1.33 | 1.26 | 1.40 | 2.19 | 1.34 | 3.59 | 1.34 | 0.85 | 2.09 | 0.41 |
| Hearing loss | 1.23 | 1.17 | 1.30 | 1.29 | 0.82 | 2.01 | 1.32 | 0.87 | 2.00 | 0.82 |
| Obesity | 1.14 | 1.07 | 1.20 | 0.98 | 0.58 | 1.65 | 0.97 | 0.63 | 1.49 | 0.40 |
| Drinking >21 units alcohol per week vs drinking 1–21 units | 0.96 | 0.89 | 1.04 | 2.67 | 1.09 | 6.56 | 1.67 | 0.70 | 3.99 | 0.22 |
| Current smoker | 1.23 | 1.12 | 1.34 | 0.90 | 0.36 | 2.27 | 1.03 | 0.44 | 2.40 | 0.58 |
| History of depression vs no reported history | 1.38 | 1.31 | 1.46 | 2.49 | 1.56 | 3.98 | 1.25 | 0.73 | 2.13 | 0.43 |
| Less than daily/almost daily social contact vs less than this | 1.14 | 1.07 | 1.22 | 1.75 | 0.95 | 3.20 | 1.30 | 0.84 | 2.00 | 0.40 |
| Physical inactivity (<WHO recommendations) | 1.59 | 1.48 | 1.71 | 1.66 | 1.00 | 2.74 | 1.20 | 0.73 | 1.98 | 0.34 |
| $PM_{2.5}$ Air pollution >WHO recommendation | 1.06 | 1.00 | 1.12 | 1.38 | 0.82 | 2.34 | 0.97 | 0.57 | 1.66 | 0.96 |
| Diabetes | 2.03 | 1.88 | 2.19 | 2.26 | 1.43 | 3.57 | 1.62 | 1.03 | 2.53 | 0.35 |
| High total cholesterol (>6.5mmol/l) | 0.86 | 0.81 | 0.93 | 0.56 | 0.20 | 1.55 | 1.20 | 0.65 | 2.23 | 0.42 |

compared with drinking 1–21 units in the White group but did find an association in the South Asian and Black groups (HR 0.96 95%CI 0.89–1.04 in White group; HR 2.68, 95%CI 1.09–6.56 in the South Asian group; HR 1.66, 95%CI 0.70–3.96 in the Black group) although confidence intervals for the latter were wide and included one. There was no evidence to support an interaction effect. In those drinking over 21 units of alcohol per week, maximum alcohol consumption was 100 units in the White group, 91 in South Asians and 95 in Black participants.

## Discussion

In this first study to compare the effect of risk factors for dementia in three different ethnicities we found no evidence to support the hypothesis that ethnicity modifies the association between any of the specified risk factors and dementia. We found evidence of a higher risk for dementia in people of self-defined Black ethnicity which remained even after adjusting for all 12 included risk factors but was eliminated when APOE status was adjusted for. In this cohort, Black participants had a higher and South Asian people a lower proportion of people who carried the APOE ε4 allele compared to White participants. This could explain the similar risk of dementia in South Asians compared to White participants even though the former have higher rates of modifiable risk factors such as hypertension and diabetes. This work suggests that in this cohort South Asian participants have a higher burden of modifiable risk factors but relatively low genetic risk whereas Black participants have a higher genetic risk as well as higher burden of modifiable risk factors. Dementia prevention efforts should therefore be more focused on these groups within the UK.

The UK Biobank has relatively good representation of minority ethnic groups compared to other UK cohorts but the number of dementia cases in minority ethnic groups was relatively low, leading to wide confidence intervals for all estimates. This is a large cohort with long follow-up and we have shown the association between ethnicity and dementia and stratified hazard ratios for 12 dementia risk factors across the three largest UK ethnic groups. To our knowledge this is the first study of the association between these key risk factors and dementia and shows no evidence of a difference in different ethnicities. In this study, people who self-identify as Black have a higher dementia risk compared to those who identify as White and South Asian but dementia outcomes are relatively rare in this generally well-educated, healthy volunteer cohort so may not be generalisable, although it is consistent with previous findings of higher rates of dementia in the Black population found in more representative cohorts [38]. The finding of a higher frequency of APOE ε4 allele in this Black population is consistent with studies of African Americans in the US but its association with dementia in these studies has been inconsistent. In some studies the allele has been found to have a lesser effect on Alzheimer's disease or cognition while in others it has been related to higher risk of dementia and this has been linked to cholesterol metabolism and cardiovascular disease and in others, to increased amyloid deposition in this population [39]. We did not find the former mechanism explained the increased risk. Mean age at leaving education was similar across all ethnic groups but the proportion of participants with education only up to age 16 was highest in the White British group. This is not likely to represent most older people from minority ethnic groups in the UK, and this may also limit generalisability, though attainment levels are rising in younger generations.

We have replicated previous findings of a protective effect of education and an increase in dementia risk from most previously studied risk factors including diabetes, hypertension, hearing loss, obesity, smoking, social isolation, physical inactivity, air pollution and depression. Within these risk factors diabetes had the highest estimated HR for dementia, yet despite

there being a much higher prevalence of this and other risk factors in South Asian participants we found no increased dementia risk in this group, possibly due to lower genetic risk. We found no clear risk of dementia from higher total cholesterol, and drinking more than 21 units of alcohol a week over long term follow up. Within the Whitehall cohort studies, it was found that abstinence from alcohol and consuming more than 14 units per week are associated with an increased risk of dementia, with dementia risk in abstainers partly accounted for by cardio-metabolic disease [40]. We did not replicate this, but alcohol consumption was only measured once compared to three different measures in the Whitehall study and the UK Biobank sample of healthy volunteers may have unmeasured confounding. Findings from neuroimaging studies suggest that any amount of alcohol results in brain damage and atrophy, including in UK Biobank participants but this does not equate to a clinical diagnosis of dementia and we did not find evidence that moderate drinking leads to a clinical problem. Equally, the finding of an increased risk of dementia for those not drinking any alcohol could be due to bias or unmeasured confounding.

The rates of high cholesterol were higher than other population based estimates in this healthy population without excess diabetes or obesity rates and it may be that the high cholesterol is due to higher levels of HDL, which is protective for cardiovascular disease and may also protect against dementia. We did not test this as HDL and LDL were not in our requested dataset as they were not part of our planned analysis.

History of help-seeking for depression was relatively high compared to previous estimates of lifetime prevalence of major depressive disorder which may partly be explained by the older sample and that it was not only major depression which was considered. The measure of depression we used showed some correlation with validated diagnoses of depression but participants were asked about any help-seeking for any level of depressive symptoms and the survey did not ascertain whether symptoms would meet clinical diagnostic thresholds. Our results show a robust association with dementia even for those who seek help from primary care, which may have relevance for population level prevention strategies.

The strengths of this study were relatively long follow-up, the large sample size and extensive data collection meaning we were able to examine all 12 risk factors and to adjust for age, sex and deprivation. The limitations are that the UK Biobank cohort is derived from volunteers and is, in general, healthier and better educated than the general population. This potential selection bias means prevalence estimates of risk factors may not be generalisable to the general population, although it is likely that the effect of risk factors are still valid. However, there is also the possibility of collider bias which could distort associations between variables and has been raised as a concern regarding the UK Biobank, among other cohort studies [41]. There were relatively few people with dementia in minority ethnic groups, so interaction analyses were underpowered. Linkage to electronic health records is not complete for all UK Biobank participants so dementia diagnosis may have been missed in some cases. Validation of dementia diagnosis have not been disaggregated by ethnic group so it is unclear if dementia diagnosis is equally valid across all ethnic groups and previous work suggests it may not be [42]. In addition, while obesity and cholesterol measures were taken and therefore well measured, depression, hearing, hypertension and diabetes were all self-reported and therefore may not be accurate. We did not measure blood pressure or take into account whether hypertension was controlled by medication or not. Risk factor status may have changed over the course of the study but we did not take this into account, for example, some people may have stopped smoking or people may have developed hypertension who did not have it at baseline. We were unable to make any distinction between first and subsequent generation migrants because the information about this was unavailable. Our analyses of inter-ethnic differences in dementia risk were unplanned but we reasoned that investigation of the observed differences in

dementia risk were important. Intra-ethnic variability is high, including for APOE distribution [43] and therefore we cannot necessarily extrapolate findings from this cohort to other minority ethnic groups within the UK or elsewhere.

Overall, our results indicate that there is no evidence to support risk factors having different associations with dementia risk in the three main ethnic groups in the UK. However, these interaction analyses are underpowered. Dementia prevention targets should be the same across these ethnic groups and particularly targeted towards minority ethnic groups who have higher rates of some risk factors. Further work is needed in a more representative population and with larger number of cases in minority ethnic groups in order to consider effect modification in more detail.

## Supporting information

**S1 File. Contains all the supporting tables.**
(DOCX)

## Acknowledgments

This research has been conducted using data from UK Biobank, a major biomedical database (www.ukbiobank.ac.uk).

## Author Contributions

**Conceptualization:** Naaheed Mukadam, Louise Marston, Gill Livingston.

**Data curation:** Naaheed Mukadam.

**Formal analysis:** Naaheed Mukadam.

**Investigation:** Naaheed Mukadam.

**Methodology:** Naaheed Mukadam, Louise Marston, Gemma Lewis, Gill Livingston.

**Project administration:** Naaheed Mukadam.

**Writing – original draft:** Naaheed Mukadam.

**Writing – review & editing:** Naaheed Mukadam, Louise Marston, Gemma Lewis, Gill Livingston.

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
