## [Decision Letter · Decision Letter 0]

14 Dec 2021

PONE-D-21-24344Risk factors, ethnicity and dementia: a UK Biobank prospective cohort study of White, South Asian and Black participantsPLOS ONE

Dear Dr. Mukadam,

Thank you for submitting your manuscript to PLOS ONE. After careful consideration, we feel that it has merit but does not fully meet PLOS ONE’s publication criteria as it currently stands. Therefore, we invite you to submit a revised version of the manuscript that addresses the points raised during the review process.

We look forward to receiving your revised manuscript.

Kind regards,

Vincenzo De Luca

Academic Editor

PLOS ONE

Journal Requirements:

Reviewers' comments:

Reviewer's Responses to Questions

**Comments to the Author**

1. Is the manuscript technically sound, and do the data support the conclusions?

Reviewer #1: Yes

2. Has the statistical analysis been performed appropriately and rigorously? 

Reviewer #1: Yes

3. Have the authors made all data underlying the findings in their manuscript fully available?

Reviewer #1: No

4. Is the manuscript presented in an intelligible fashion and written in standard English?

Reviewer #1: Yes

5. Review Comments to the Author

Reviewer #1: If the biobank data on subjects could be provided by omitting subject's name or other sensitive information, further statistical exploration or meta-analysis of the data could be encouraged along with transparency around findings that comes with it.

6. PLOS authors have the option to publish the peer review history of their article (what does this mean?). If published, this will include your full peer review and any attached files.

Reviewer #1: **Yes: **Arojit Mitra

---

## [Author Response · Author response to Decision Letter 0]

25 Jan 2022

In this paper, authors have leveraged the rich UK Biobank data to parse out similarity and

dissimilarity, if any, of risk factors for dementia among different ethnic populations of the UK.

The paper is extremely well written using reasonable statistical methods.

- Thank you for your positive comments

I have following minor comments/suggestions for the authors to consider:

1. The paper is heavy at the data side and therefore, a pictorial presentation of their

cohort diversity and risk factors could aid in readability. Authors could consider pie

charts to show the diversity (number/age/sex etc) and characteristic of their samples.

- We considered pictorial representation but concluded that as there are three main ethnic groups and these are manageable within a table, that this would be easily understood and assimilated by readers.

2. The two ethnic minorities, Black and South Asians, are certainly underpowered (less

than 2 % of the total subject) as rightly pointed out by the authors. Wondering if

resampling the White population in this study using bootstrapping would still lead to

similar statistical outcome or there is heterogeneity in the White population that is

getting masked by oversampling?

- Although the percentage of people from minority ethnic groups was relatively small, total number was over 2700 for both groups. Ethnicity data was relatively complete so we did not think it appropriate to use statistical techniques to make assumptions about ethnicity as this could bias the results.

3. Can the authors also calculate and compare the effect size of the contribution of the

risk factors leading to dementia in the three ethnic populations?

- This is a good suggestion but was not part of the aims of this paper. 

4. If I am not wrong, UK Biobank does collect family history that can be helpful to know

if the parents or grandparents suffered from dementia. Wondering if this information

is available for the studied cohort? If yes, it will be good to factor it in and see if it

could be considered as a predictive factor.

- Yes UK Biobank does collect family history but as this is not a potentially modifiable risk factor we did not include this in our analyses.

5. The main results of the paper are all buried in text. A correlation matrix using heat

maps or similar visual representations would be very helpful.

- We are not sure how a correlation matrix could be used to illustrate our findings. We have used tables wherever possible, to avoid over-writing the results.

6. In table 1, row 3, ‘Hypertension’, was confirmed by either self-report or medication

history. In the discussion section, author’s have mentioned the following

‘...depression, hearing, hypertension and diabetes were all self-reported and

therefore may not be accurate’. This is confusing. If medication records are available,

why not limit the observation to those subjects excluding the

self-diagnosed/self-reported cases?

- We agree this is misleading. We have now revised it to say that hypertension was either through self-reported diagnosis or reported use of hypertensive medications. As both of these are self-report measures and people are more likely to under-report than over-report their medical conditions, we defined people with hypertension as those who reported either or both.

7. In the ‘Analysis of association of risk factors with dementia-Ethnicity’ section, the

term ‘Asian Bangladeshi’ is confusing. What is the difference between Asain

Bangladeshi and Bangladeshi?

- We have now removed the word “Asian” from this line so it now reads “South Asian (Bangladeshi, Indian, Pakistani)”

8. In this study, participants aged 55 and over were chosen with a follow-up period of 15

years. So the age range of the sample is 55-70 years. It seems that the sample could

be further characterized into early- (subjects with age below 65 years) and late-onset

dementia. The difference between the risk factors between these two types should

be clinically relevant, elevating the impact of the paper.

- We considered doing this but the majority of cases are late onset and if we separate cases into late and early onset then stratifying by ethnic group results in very small sample sizes for the analyses.

---

## [Decision Letter · Decision Letter 1]

10 Jul 2022

PONE-D-21-24344R1Risk factors, ethnicity and dementia: a UK Biobank prospective cohort study of White, South Asian and Black participantsPLOS ONE

Dear Dr. Mukadam,

Thank you for submitting your manuscript to PLOS ONE. After careful consideration, we feel that it has merit but does not fully meet PLOS ONE’s publication criteria as it currently stands. Therefore, we invite you to submit a revised version of the manuscript that addresses the points raised during the review process. Your manuscript has been assessed by three reviewers who are supportive about your study. To ensure that your submission meets our publication criteria (https://journals.plos.org/plosone/s/criteria-for-publication#loc-3;
https://journals.plos.org/plosone/s/criteria-for-publication#loc-4), please address all the comments noted below.

We look forward to receiving your revised manuscript.

Kind regards,

Alejandra Clark

Division Editor

PLOS ONE

Journal Requirements:

Reviewers' comments:

Reviewer's Responses to Questions

**Comments to the Author**

1. If the authors have adequately addressed your comments raised in a previous round of review and you feel that this manuscript is now acceptable for publication, you may indicate that here to bypass the “Comments to the Author” section, enter your conflict of interest statement in the “Confidential to Editor” section, and submit your "Accept" recommendation.

Reviewer #1: All comments have been addressed

Reviewer #2: All comments have been addressed

Reviewer #3: All comments have been addressed

2. Is the manuscript technically sound, and do the data support the conclusions?

Reviewer #1: Yes

Reviewer #2: Yes

Reviewer #3: Yes

3. Has the statistical analysis been performed appropriately and rigorously? 

Reviewer #1: Yes

Reviewer #2: Yes

Reviewer #3: Yes

4. Have the authors made all data underlying the findings in their manuscript fully available?

Reviewer #1: No

Reviewer #2: No

Reviewer #3: Yes

5. Is the manuscript presented in an intelligible fashion and written in standard English?

Reviewer #1: Yes

Reviewer #2: Yes

Reviewer #3: Yes

6. Review Comments to the Author

Reviewer #1: (No Response)

Reviewer #2: This study was analysed and written very well. Although there are various factors about ethnicity and dementia risks left to explore, the study utilized data very well and made a tangible conclusion. In addition, a Kaplan-Meyer plot of the dementia conversion rate on the three ethics would help readers to get information more clearly. It would be beneficial to add some suggestions for future research on how to collect data that should be done to disentangle the ethnic issues on dementia. My two comments are merely soft suggestions, the author can choose to correction or not.

Reviewer #3: The study is an interesting study that investigates the risk factors, ethnicity, and dementia. The result of APOE in these 3 subgroups of the population was also interesting. I have some comments as follows.

1. The diagnosis of dementia which was established through self-report or linked electronic health records in the UK Biobank using a defined algorithm and recognised ICD9 and ICD10 codes. I am wondering if the self-report contained high accuracy or not, especially for those in an ambulatory care or community care who had not been hospitalized or dead where the accuracy might be higher.

2. The 2 ethnic groups were relatively small in proportion (3590 South Asian and 2766 Black people). I could understand that it would be a normal distribution of such a proportion in the general population in the UK, but the incidence of dementia and rare risk factors might be less precised and provided wide confidence interval.

3. The risk factors might be defined at the beginning of data collection. I am not sure if the risk factors could change over time such as from current smoke to non-smoke. Would this lead to potential misclassification of the risk factors.

4. I am wondered if the other illnesses which might develop later after the study entry and could increase the risk of dementia were collected and analysed.

7. PLOS authors have the option to publish the peer review history of their article (what does this mean?). If published, this will include your full peer review and any attached files.

Reviewer #1: **Yes: **Arojit Mitra

Reviewer #2: **Yes: **Chavit Tunvirachaisakul

Reviewer #3: **Yes: **Professor Weerasak Muangpaisan

---

## [Author Response · Author response to Decision Letter 1]

12 Jul 2022

Reviewer #2: This study was analysed and written very well. Although there are various factors about ethnicity and dementia risks left to explore, the study utilized data very well and made a tangible conclusion. In addition, a Kaplan-Meyer plot of the dementia conversion rate on the three ethics would help readers to get information more clearly. It would be beneficial to add some suggestions for future research on how to collect data that should be done to disentangle the ethnic issues on dementia. My two comments are merely soft suggestions, the author can choose to correction or not.

- Thank you for these suggestions. As the number of dementia cases in the minority ethnic groups is so low we did not think we would have sufficient data points for a Kaplan Meier plot. We have suggested in our conclusions that future studies use a larger sample of people from minority ethnic groups and a more representative population to disentangle these questions further.

Reviewer #3: The study is an interesting study that investigates the risk factors, ethnicity, and dementia. The result of APOE in these 3 subgroups of the population was also interesting. I have some comments as follows.

1. The diagnosis of dementia which was established through self-report or linked electronic health records in the UK Biobank using a defined algorithm and recognised ICD9 and ICD10 codes. I am wondering if the self-report contained high accuracy or not, especially for those in an ambulatory care or community care who had not been hospitalized or dead where the accuracy might be higher.

- We were not able to ascertain the validity of dementia diagnosis in this study but previous studies using the UK Biobank have shown a high validity for dementia diagnosis and we have cited this literature in our introduction. 

2. The 2 ethnic groups were relatively small in proportion (3590 South Asian and 2766 Black people). I could understand that it would be a normal distribution of such a proportion in the general population in the UK, but the incidence of dementia and rare risk factors might be less precised and provided wide confidence interval.

- We agree the sample sizes are relatively small and have mentioned this in our limitations.

3. The risk factors might be defined at the beginning of data collection. I am not sure if the risk factors could change over time such as from current smoke to non-smoke. Would this lead to potential misclassification of the risk factors.

- We agree the risk factor classification could change over time and have now added this sentence to our discussion: “Risk factor status may have changed over the course of the study but we did not take this into account, for example, some people may have stopped smoking or people may have developed hypertension who did not have it at baseline.”

4. I am wondered if the other illnesses which might develop later after the study entry and could increase the risk of dementia were collected and analysed.

- We have addressed this point with the sentence copied above – as we only measured risks at baseline we may have missed some.

---

## [Decision Letter · Decision Letter 2]

14 Sep 2022

Risk factors, ethnicity and dementia: a UK Biobank prospective cohort study of White, South Asian and Black participants

PONE-D-21-24344R2

Dear Dr. Mukadam,

We’re pleased to inform you that your manuscript has been judged scientifically suitable for publication and will be formally accepted for publication once it meets all outstanding technical requirements.

Kind regards,

Gyaneshwer Chaubey

Academic Editor

PLOS ONE

Additional Editor Comments (optional):

Reviewers' comments:

Reviewer's Responses to Questions

**Comments to the Author**

1. If the authors have adequately addressed your comments raised in a previous round of review and you feel that this manuscript is now acceptable for publication, you may indicate that here to bypass the “Comments to the Author” section, enter your conflict of interest statement in the “Confidential to Editor” section, and submit your "Accept" recommendation.

Reviewer #1: All comments have been addressed

Reviewer #2: All comments have been addressed

Reviewer #3: All comments have been addressed

2. Is the manuscript technically sound, and do the data support the conclusions?

Reviewer #1: Yes

Reviewer #2: Yes

Reviewer #3: Yes

3. Has the statistical analysis been performed appropriately and rigorously? 

Reviewer #1: Yes

Reviewer #2: Yes

Reviewer #3: Yes

4. Have the authors made all data underlying the findings in their manuscript fully available?

Reviewer #1: Yes

Reviewer #2: Yes

Reviewer #3: Yes

5. Is the manuscript presented in an intelligible fashion and written in standard English?

Reviewer #1: Yes

Reviewer #2: Yes

Reviewer #3: Yes

6. Review Comments to the Author

Reviewer #1: (No Response)

Reviewer #2: The authors answered and clarified all the noted points. Thank you for producing a very good research.

Reviewer #3: The study is an interesting study that investigates the risk factors, ethnicity, and dementia. Though, there are some limitations of the study, the authors have addressed these issues in the discussion part.

7. PLOS authors have the option to publish the peer review history of their article (what does this mean?). If published, this will include your full peer review and any attached files.

Reviewer #1: **Yes: **Arojit Mitra

Reviewer #2: **Yes: **Chavit Tunvirachaisakul, MD. Ph.D.

Reviewer #3: **Yes: **Professor Weerasak Muangpaisan

---

## [Editor Report · Acceptance letter]

20 Sep 2022

PONE-D-21-24344R2 

Risk factors, ethnicity and dementia: a UK Biobank prospective cohort study of White, South Asian and Black participants 

Dear Dr. Mukadam:

I'm pleased to inform you that your manuscript has been deemed suitable for publication in PLOS ONE. Congratulations! Your manuscript is now with our production department. 

Kind regards, 

on behalf of

Gyaneshwer Chaubey 

Academic Editor

PLOS ONE